# Enhanced Gaussian Process Model for Predicting Compressive Strength of Ultra-High-Performance Concrete (UHPC)

**DOI:** 10.3390/ma17246140

**Published:** 2024-12-16

**Authors:** Zhipeng Zou, Bin Peng, Lianghai Xie, Shaoxun Song

**Affiliations:** School of Environment and Architecture, University of Shanghai for Science and Technology, Shanghai 200093, China; 222271906@st.usst.edu.cn (Z.Z.); 223382069@st.usst.edu.cn (L.X.); 2035053417@st.usst.edu.cn (S.S.)

**Keywords:** ultra-high-performance concrete, compressive strength, mix proportions, Gaussian process, singular value decomposition, Kalman filtering and smoothing

## Abstract

Ultra-high-performance concrete (UHPC) is widely used in engineering due to its exceptional mechanical properties, particularly compressive strength. Accurate prediction of the compressive strength is critical for optimizing mix proportions but remains challenging due to data dispersion, limited data availability, and complex material interactions. This study enhances the Gaussian Process (GP) model to address these challenges by incorporating Singular Value Decomposition (SVD) and Kalman Filtering and Smoothing (KF/KS). SVD improves data quality by extracting critical features, while KF/KS reduces data dispersion and align prediction with physical laws. The enhanced GP model predicts compressive strength with improved accuracy and quantifies uncertainty, offering significant advantages over traditional methods. The results demonstrate that the enhanced GP model outperforms other models, including artificial neural networks (ANN) and regression models, in terms of reliability and interpretability. This approach provides a robust tool for optimizing UHPC mix designs, reducing experimental costs, and ensuring structural performance.

## 1. Introduction

Ultra-high-performance concrete (UHPC) is characterized by its high strength, low porosity, and favorable ductility compared to normal concrete. With these advantages, it is increasingly applied in engineering practice [1,2,3]. The compressive strength is one of the most fundamental mechanical properties of UHPC, which is critically influenced by the mix proportions, such as cement content, silica fume, fly ash, and the water-to-binder ratio. The water-to-binder ratio affects the hydration process, impacting strength development, while silica fume and fly ash refine the concrete microstructure, enhancing density and strength. Accurately predicting compressive strength based on these mix proportions is essential for optimizing the performance of UHPC. However, the prediction is challenging due to the complexity of the interactions among these components and their influence on strength.

Experimental methods can determine the relationship between mix proportions and compressive strength. Wang and Gao [4] experimentally investigated how varying steel fiber contents affect the flowability and compressive strength of UHPC, finding that higher fiber content reduced air entrapment and increased strength. Zhao et al. [5] demonstrated through experiments that vibratory mixing significantly enhanced the unconfined compressive strength under different vibration accelerations and cement dosages. Pourbaba et al. [6] conducted compressive tests on 180 specimens of UHPC at various ages and found that it reached over 90% of its final strength within 21 days. Arabyarmohammadi et al. [7] used response surface methodology combined with experiments to optimize the mix proportions of roller-compacted concrete (RCC) and revealed that cement content had a significant impact on compressive strength. Xiao et al. [8] studied the effect of different steel fiber contents on the compressive properties of ultra-high-performance shotcrete (UHPSC) through axial compression experiments, showing that higher fiber content improved strength and toughness. However, curing and testing consume too much time and yield limited results.

Deterministic models have been widely used to predict the strength of concrete. Barkhordari et al. [9] applied ensemble neural networks to predict the compressive strength of fly ash concrete, achieving high accuracy through methods like stacking and bagging. Huang et al. [10] combined Random Forest (RF) and Firefly algorithms to predict compressive strength in metakaolin cement-based materials, emphasizing the importance of cement grade and water-to-binder ratios. Jaf et al. [11] proposed soft computing models, including artificial neural network (ANN) and regression models, to predict the slump and compressive strength of self-compacted concrete modified with fly ash. Li et al. [12] used gradient boosting regression trees (GBRT) to predict concrete compressive strength, achieving high accuracy with a 0.92 determination coefficient. Vargas et al. [13] demonstrated the integration of machine learning models, such as Extreme Gradient Boosting Regressor (XGBoost) and neural networks, to optimize compressive and flexural strength in concrete design, reducing costs and enhancing reliability. Recent studies have demonstrated the potential of machine learning approaches in advancing UHPC-related research and concrete optimization. For example, interpretable machine learning models have been used to predict UHPC creep behavior, showcasing their ability to balance accuracy and transparency while addressing complex mechanical behaviors [14]. Similarly, machine learning approaches combined with optimization models have been applied to evaluate eco-efficiency, enabling sustainable design in concrete applications [15].

However, these models cannot represent the uncertainty of the compressive strength of the UHPC [16,17,18]. The inherent uncertainty results from the variability of ingredient in the mix, diversity of curing environment, difference in hydration process, and so on.

To address these limitations, the Gaussian process (GP) model has potential advantages. First, as a nonparametric model, the GP model does not rely on complex equations or structures, so the complexity of hyperparameter tuning decreases. Second, the GP model can use prior knowledge, including plausible physical law, as mean function. According to the Bayesian principles, it improves the prediction even with small datasets and enhances the interpretability [19,20,21]. Third, the GP model demonstrates high flexibility by using different kernel functions [22,23]. Fourth, the GP model quantifies uncertainty by providing probability distributions with its prediction. The GP model has demonstrated its advantages in predicting the strength of high-performance concrete (HPC) and lightweight aggregate concrete [24,25].

The GP model can be directly applied to the design of UHPC by predicting the compressive strength of new mix proportions without the need for physical testing. It can also provide predicted compressive strength values along with uncertainty estimates. This approach enables rapid evaluation and optimization of mix proportion designs, significantly reducing the time and cost of experiments while ensuring structural requirements are met in the early design stages.

Considering the advantages of the GP model, this study seeks to apply it to predict the compressive strength of UHPC. The structure of the paper is as follows: Section 2 introduces the methodology. Section 3 introduces the datasets used to train and test the model. Section 4 describes the development and application of the GP model, covering its theoretical foundations, training process, hyperparameter optimization, and performance evaluation. Section 5 focuses on improving the GP model through data augmentation techniques, such as merging heterogeneous datasets using Singular Value Decomposition (SVD) and reducing data dispersion using Kalman Filtering and Smoothing (KF/KS). Section 6 provides a comprehensive comparison of the GP model’s predictions with alternative models, such as artificial neural networks (ANN) and regression-based methods, evaluating their relative accuracy and reliability. Finally, Section 7 concludes the study by summarizing the findings, highlighting the contributions, and discussing potential directions for future research. Figure 1 provides an overview of the research methodology, outlining the key steps from data collection and preprocessing to model training, enhancement, and final prediction.

Predicting the compressive strength of UHPC remains challenging due to the high variability of material properties and the limited availability of experimental data. Traditional methods often struggle to achieve a balance between accuracy, interpretability, and reliability. To address these limitations, this study presents three key contributions. (1) An enhanced Gaussian Process (GP) model is proposed, incorporating Singular Value Decomposition (SVD) and Kalman Filtering/Smoothing (KF/KS) to improve the prediction accuracy by increasing the volume of data and reducing data dispersion. (2) The GP model utilizes prior knowledge, including physical laws, to enhance prediction performance. (3) The model provides quantification of prediction uncertainty, which is advantageous for reliability assessment in engineering applications. These contributions aim to address the challenges in compressive strength prediction for UHPC, offering an efficient and reliable solution for optimizing UHPC mix designs.

## 2. Methodology

In this section, we present the methodology adopted in this research to predict the compressive strength of ultra-high-performance concrete (UHPC). The research objectives are to (1) develop an enhanced GP model that accurately predicts compressive strength based on mix proportions, (2) incorporate data augmentation techniques such as SVD and KF/KS to improve model reliability, and (3) provide uncertainty quantification to support practical decision-making. The scope of this research encompasses data preprocessing, model training, and evaluation using experimental datasets of UHPC with various mix compositions. The methodology involves collecting datasets, preprocessing using SVD to manage data heterogeneity, and using KF/KS to improve training data quality. The enhanced GP model is trained and validated against other predictive models, such as ANN and Polynomial Regression (PR), to demonstrate its effectiveness in compressive strength prediction.

## 3. Datasets of the Compressive Strength of UHPC

The data in this study were our experimental results of the compressive strength of UHPC and other experimental results available in the literature [26,27]. These data cover a broad range of UHPC mixtures and compressive strength values.

Dataset 1 consists of experimental results obtained in this study, whereas the literature results were divided into Dataset 2 and Dataset 3. In all three datasets, the output variable was the 28-day compressive strength, and the input variables varied to fully account for the influence of various mixtures, as shown in Table 1. Figure 2 shows the distribution of the compressive strength. Appendix A provide detailed information on the datasets.

In this study, Min–Max normalization was used to preprocess the input variables before training the model. This approach scales each feature to a range of [0, 1], preserving the relative relationships within the data, which helps manage different variable scales effectively.
(1)$$\bar{x}=\frac{x-\min\left(x\right)}{\max\left(x\right)-\min\left(x\right)}$$
where $\bar{x}$ and x are the normalized and original value of a sample of an input variable, respectively;$\;\max\left(x\right)$ and $\min\left(x\right)$ are the maximum and minimum value of the input variable, respectively.

The normalized data were divided into two parts: 80% for the training set and the remaining 20% for the test set.

## 4. Prediction of the Compressive Strength Based on GP

### 4.1. Basic Principles of the GP Model

The GP model to predict the compressive strength of UHPC can be expressed as follows:(2)$$f\left(x\right)∼GP\left(m\left(x\right),K\left(x,x^{\prime }\right)\right)$$
where $f\left(x\right)$ is the compressive strength of UHPC; x is the column vector of input variables; $m\left(x\right)$ is the mean function of the compressive strength; $K(x,x^{\prime })$ is the covariance matrix of the compressive strength; GP denotes the Gaussian process.

The process for making predictions involves several key steps:Step 1: Model Definition

The GP model is characterized by its mean function and kernel function. The mean function can take forms such as a constant function, a polynomial regression equation, or any other plausible physical laws. It incorporates the prior knowledge about the compressive strength. With the prior knowledge, the GP model is more interpretable and less dependent on the experimental results. The kernel function can be selected from a big family, including the squared exponential functions, Matern 5/2 functions, rational quadratic functions, and so on. It makes the GP model flexible. Appropriately selecting the mean and kernel functions before training the model can apply prior judgments about the compressive strength and improve the efficiency of the training.

Step 2: Model Training

The GP model is trained using available experimental data to determine the optimal hyperparameters. This is achieved by maximizing the following log-likelihood function:(3)$$\mathrm{Log}p\left(f|x,\theta \right)=-\frac{1}{2}f^{T}K^{-1}f-\frac{1}{2}\log det\left(K\right)-\frac{n}{2}\log2\pi$$
where $\theta ={\left(\sigma_{f},l,\sigma_{n},\alpha \right)}^{T}$ is the column vector of hyperparameters introduced by the prior model; $K=K(x,x^{\prime })$ is the covariance matrix; n is the number of experimental results, i.e., training data points. In Equation (3), the first term on the right-hand side represents how well the experimental results are fitted; the second term is a penalty term that helps prevent overfitting or using overly complex models; the third term is a normalization constant [28].

During training, the GP model aims to find the hyperparameters that best describe the observed data while balancing model complexity and avoiding overfitting. The training process combines prior judgment information with information from the existing data in a weighted manner.

Step 3: Setting up the joint distribution

To predict the compressive strength for a new input data point $x^{*}$, it is assumed that the corresponding output $f\left(x^{*}\right)$ and $f\left(x\right)$ follow a joint normal distribution; thus:(4)$$\left(\begin{array}{c} f\left(x\right)\\ f\left(x^{*}\right) \end{array}\right)∼N\left(\left[\begin{array}{c} m\left(x\right)\\ m\left(x^{*}\right) \end{array}\right],\left[\begin{array}{cc} K\left(x,x\right) & K\left(x,x^{*}\right)\\ K\left(x^{*},x\right) & K\left(x^{*},x^{*}\right) \end{array}\right]\right)$$
where N denotes the joint normal distribution; K(x,x) is the covariance matrix of the training data; $K(x,x^{*})$ is the covariance matrix between training data and new input data; $K(x^{*},x^{*})$ is the covariance matrix of the new input data itself. This joint normal distribution forms the basis for predicting the new output based on the existing data.

Step 4: Calculation of posterior mean and variance

The $f\left(x^{*}\right)$ can be expressed in terms of the hyperparameters of the trained model as follows:(5)$$f\left(x^{*}|x\right)∼N\left(\mu_{f},\sigma_{f}^{2}\right)$$
where $\mu_{f}$ is the posterior mean; $\sigma_{f}^{2}$ is the posterior variance. The posterior mean and posterior variance are expressed as follows:(6)$$\mu_{f}=m\left(x^{*}\right)+K\left(x^{*},x\right){K\left(x,x\right)}^{-1}\left(\left[f\left(x\right)-m\left(x\right)\right]\right)$$
(7)$$\sigma_{f}^{2}=K\left(x^{*},x^{*}\right)-K\left(x^{*},x\right){K\left(x,x\right)}^{-1}K\left(x,x^{*}\right)$$

The posterior mean represents the expected value of the compressive strength for the new input. This value is influenced by the deviations observed in the training data, which are weighted by the covariance between the new input and the training points. The posterior variance quantifies the uncertainty in the prediction. A lower variance indicates a higher confidence in the predicted value.

Step 5: Generating the prediction

The final prediction for the new input is represented as a normal distribution characterized by the calculated posterior mean and posterior variance. This means that the predicted compressive strength is not just a point estimate but also includes a measure of uncertainty. This is crucial for practical engineering applications where understanding both the expected performance and associated risks is essential.

Step 6: Updating the model

The GP model is highly adaptable and can be updated when new data becomes available. When additional data points are introduced, the posterior mean and variance can be recalculated to refine the model’s predictions. This continuous learning process ensures that the GP model remains accurate and relevant, providing reliable predictions even as new information is acquired.

### 4.2. Prediction of the Compressive Strength

The mean of the training set was selected as the prior mean function for the GP model. This choice provides a reasonable initial prediction and reflects the trend of the compressive strength of UHPC. We selected the Matern 5/2 kernel function due to its flexibility in handling variations across different scales, which is crucial for UHPC data characterized by heterogeneous and non-linear relationships. Compared to other potential kernels, such as the Radial Basis Function (RBF) kernel or the Matern 3/2 kernel, the Matern 5/2 kernel strikes an ideal balance between smoothness and adaptability. The RBF kernel, while providing very smooth predictions, can be too restrictive and over-smooth complex real-world data, potentially missing important local variations. On the other hand, the Matern 3/2 kernel offers less smoothness but does not capture the moderate variability as effectively as the Matern 5/2 kernel. Empirical testing demonstrated that the Matern 5/2 kernel consistently provided better predictive accuracy and generalization capability, making it the optimal choice for modeling the compressive strength of UHPC.

Five-fold cross-validation was performed on the training set to determine the optimal hyperparameters. During each cross-validation iteration, the performance metrics on the validation set were recorded, including the goodness-of-fit (denoted by *R*^2^) and root-mean-square error (RMSE). *R*^2^ measures the consistency of the predicted values with the actual trend, and RMSE measures the overall deviation between predicted and actual values. The two metrics provide a comprehensive evaluation of the predictive performance. The hyperparameters resulting in best metrics were selected as the final hyperparameters for the model.

We evaluated the performance of the GP model using the three datasets. The results are shown in Figure 3. The difference among prediction of the three datasets reveals that volume and dispersion of data are critical. The prediction of dataset 3 is better than that of dataset 2 because the former has bigger data volume. The prediction of dataset 2 is better than that of dataset 1 because the former has less dispersion.

Therefore, to optimize the model, both sufficient data volume and reduced data dispersion are necessary.

## 5. Improving the GP Model

### 5.1. Merging Datasets and Extracting Information Using SVD

The available experimental results of the compressive strength of UHPC are insufficient to build a robust predictive model. Meanwhile, the experimental methods, ingredients, and mixing vary. This variation results in datasets that cannot be simultaneously used to train the GP model. To increase the volume of training data, we merged the different experimental results and applied SVD to preprocess the merged datasets.

SVD extracts the principal components of the data, effectively filtering out less informative variations, while retaining the most critical features needed for accurate prediction. By simplifying the data while preserving its most essential features, SVD ensures that the GP model is trained on high-quality, well-structured data, ultimately enhancing the model’s stability and reliability.

Table 2 illustrates the merging process. Input variables considered in all datasets are aggregated. If a dataset lacks a specific variable, which is considered in other datasets, a zero-value column is filled in the corresponding position. In this way, the datasets were expanded to have equal numbers of columns (i.e., input variables). Then, the expanded datasets are concatenated row-wise to form an input matrix.

Currently, we opted for zero-value imputation as an initial approach to ensure consistency across the combined datasets and facilitate the Singular Value Decomposition process. We recognize that this may not be the most optimal approach in all cases. In future work, we plan to improve our handling of missing data by considering alternative imputation methods, such as using mean or median values, as well as exploring advanced techniques like matrix factorization and machine learning-based imputation. These improvements will aim to enhance the reliability of the model and ensure that the impact of missing data is adequately addressed.

Then, SVD was performed on the input matrix. The number of singular values to retain was determined using the Cumulative Energy Contribution (CEC) method [29], it requires the following:(8)$$\frac{\sum_{i=1}^{m}\sigma_{i}^{2}}{\sum_{i=1}^{n}\sigma_{i}^{2}}\times 100\%\geq 95\%$$
where n is the total number of singular values; *m* is the maximum index of the singular values to be retained.

We calculated the value of *m* to be 7. Therefore, the first seven singular values of the input matrix were retained and combined with the corresponding left and right singular vectors. The combination reconstructed an input matrix. This reconstructed input matrix preserves the primary information from all datasets and provides more training data for the GP model.

Figure 4 shows the prediction using the reconstructed input matrix. Compared to Figure 3, it shows an obvious improvement in predictive performance.

### 5.2. Training Data Preprocessing Based on KF/KS

To reduce the impact of data dispersion and reinforce the role of the physical model, the data in the reconstructed input matrix from Section 5.1 was processed using KF/KS.

KF integrates prior physical knowledge directly into the predictive process to ensure the GP model’s prediction align with established physical principles. This step reduces data dispersion and refines the estimates by continuously updating predictions based on real-time observations. KS complements KF by applying a backward correction process to refine these predictions retrospectively, thus reducing residual errors and increasing overall consistency.

Both KF and KS are based on Bayesian principles and fully utilize the available information to reduce the data dispersion. The process is as follows [30]:●Calculate the prior prediction of the compressive strength for the current mix proportions:
(9)$$f_{\left[t\right]}^{*}=F{\hat{f}}_{\left[t-1\right]}$$
where ${\hat{f}}_{\left[t-1\right]}$ is the posterior estimate of the compressive strength for the previous mix proportions; $f_{\left[t\right]}^{*}$ is the prior prediction of the compressive strength for the current mix proportions; F is the state transition matrix, which represents the ratio of the calculated results based on Abram’s formula for the current and previous mix proportions [31]:(10)$$f=\alpha_{a}X^{-\alpha_{b}}$$
where f is the compressive strength of UHPC under specific mix proportions. X is the water-to-binder ratio; $\alpha_{a},\alpha_{b}$ are regression coefficients.

Abram’s formula describes the changes in compressive strength under different mix proportions. It’s a plausible physical law and prior knowledge about the compressive strength. Therefore, the state transition matrix F incorporates the prior knowledge. It is not solely data-driven but enforces the physical law.

●Calculate the covariance of measure noise for the current mix proportions:(11)$$P_{\left[t\right]}^{*}=F{\hat{P}}_{\left[t-1\right]}F^{T}$$
where ${\hat{P}}_{\left[t-1\right]}$ is the posterior covariance of measure noise for the previous mix proportions; $P_{\left[t\right]}^{*}$ is the covariance of measure noise for the current mix proportions.

●Calculate the Kalman gain for the current mix proportions:(12)$$K_{t}=P_{\left[t\right]}^{*}{\left(P_{\left[t\right]}^{*}+Q\right)}^{-1}$$
where Q is the covariance of process noise.

●Update the prior prediction in Step 1 to the posterior one:(13)$${\hat{f}}_{\left[t\right]}=f_{\left[t\right]}^{*}+K_{t}\left(f_{\left[t\right]}-f_{\left[t\right]}^{*}\right)$$
where ${\hat{f}}_{\left[t\right]}$ is the posterior prediction of the compressive strength, $f_{\left[t\right]}$ is the measured compressive strength for the current mix proportions. Since the $f_{\left[t\right]}^{*}$ incorporate prior knowledge and physical law, the update is fully Bayesian.


●Update the covariance of measure noise for the current mix proportions:

(14)
$${\hat{P}}_{\left[t\right]}=\left(1-K_{t}\right)P_{\left[t\right]}^{*}$$

●Return to step 1 and continue the calculation.


The above describes the forward recursion process of KF. The KS begins with the final estimates ${\hat{f}}_{\left[t\right]}$ and ${\hat{P}}_{\left[t\right]}$ from the KF and recursively proceeds backward to the first observation [32].

After applying KF/KS to the reconstructed input matrix from Section 5.1, we trained the GP model again. The resulting prediction Figure 5 indicates improvements compared to Figure 3 and Figure 4. Processing with KF/KS reduces the impact of dispersion in the training data. Additionally, it incorporates the physical law and enhances the reliability and interpretability of the prediction.

Using KF/KS to reduce data dispersion might limit the interpretation of natural variability, particularly for datasets characterized by high heterogeneity. We carefully chose KF/KS as these techniques effectively reduce measurement noise. This step aimed to provide more stable and reliable predictions. We understand the importance of retaining the natural variability in data, and in future studies, we plan to conduct detailed sensitivity analyses to evaluate the impact of KF/KS more thoroughly.

The GP model provide confidence intervals and then quantify the uncertainty of its prediction. It offers a more reliable basis for decision-making in engineering applications. For example, data points whose experimental values fall outside the 95% confidence interval of the predicted values can be identified according to the GP prediction. These points exhibit high uncertainty or low reliability. Therefore, they can be reasonably removed and the remaining data can further improve the prediction, as shown in see Figure 6. This approach may introduce bias by excluding potentially valid data points, which could limit the model’s ability to generalize well to new datasets. If feasible in future work, we aim to explore alternative approaches, such as using a weighted training scheme where data points with higher uncertainty are retained but assigned lower weights. This could allow us to better preserve data variability while minimizing the negative impact of noisy data on the model’s performance. However, given the current scope of this research, the focus was on ensuring model reliability and mitigating noise effects.

## 6. Qualification of the Prediction Based on GP

To comprehensively evaluate the performance of the proposed GP model, we compare its prediction with that of other methods using different data sets.

Polynomial regression (PR) and artificial neural network (ANN) are extensively used in prediction of material properties. We used them to predict the compressive strength of merged Dataset. The structure of the ANN is illustrated in Figure 7 and the polynomial is as follows:(15)$$f=\beta_{0}+\sum_{i=1}^{11}\beta_{i}x_{i}+\sum_{i=1}^{11}\beta_{ii}x_{i}^{2}+\sum_{i=1}^{11}\sum_{j=i+1}^{11}\beta_{ij}x_{i}x_{j}+\epsilon$$
where f is the column vector of the compressive strength of UHPC; x is the column vector of input variables; $\beta_{0},\beta_{i},\beta_{ii}\;and\;\beta_{ij}$ are the regression coefficients; ϵ is the error term. We compare their results and the prediction of GP in Figure 8. It should be pointed out that ANN took 2537.40 s to complete the prediction, whereas the proposed method required only 459.30 s.

In 2023, Wakjira and colleagues used various machine learning methods to predict the compressive strength of Dataset 2, including DT (decision trees), AdaBoost (adaptive boosting), GBM (gradient boosting machine), XGB (extreme gradient boosting), and SL (super-learner). We compare their results and the prediction of GP in Figure 9.

In 2020, Marani and colleagues predicted the compressive strength of Dataset 1 using RF (Random Forest), ET (extra trees), and GB (gradient boosting). We compare their results and the prediction of GP in Figure 10.

The proposed method has favorable generalizability and it achieves similar, or slightly better, prediction under all cases. In addition, it quantifies the uncertainty of the prediction and provides advantages in practices.

## 7. Conclusions

To address the challenges in compressive strength prediction for UHPC, we collected and consolidated existing experimental data to train a GP model, predicted the compressive strength of UHPC, and validate the predictive results. The research findings are as follows:(1)The GP model can well predict the compressive strength of UHPC. It utilizes prior knowledge, including physical laws, to improve the predictive performance.(2)By merging the datasets and using SVD to extract effective information, different experimental results can be fully utilized to increase the volume of training data for the GP model and improve its predictive performance.(3)Preprocessing using KF/KS reduce the adverse effects of data dispersion and enhances the reliability of the compressive strength prediction.(4)The GP model quantifies the uncertainty in its prediction and consequently offers a more reliable basis for decision-making in engineering applications.(5)The GP model has favorable generalizability.(6)The enhanced GP model allows managers to quickly evaluate mix proportions, reducing costs and time associated with physical testing. It also provides uncertainty quantification, which helps in making informed decisions for quality control and risk management in UHPC production.(7)It is important to recognize certain limitations. The exclusion of data points with high uncertainty may introduce bias, potentially affecting generalizability. Future work should explore weighting schemes instead of removing uncertain data entirely, to better preserve data diversity. Furthermore, the choice of the Matern 5/2 kernel for the GP model was effective for the given dataset, but alternative kernels should be explored in future research to confirm the robustness of this choice across various data scenarios.

## Figures and Tables

**Figure 1 materials-17-06140-f001:**
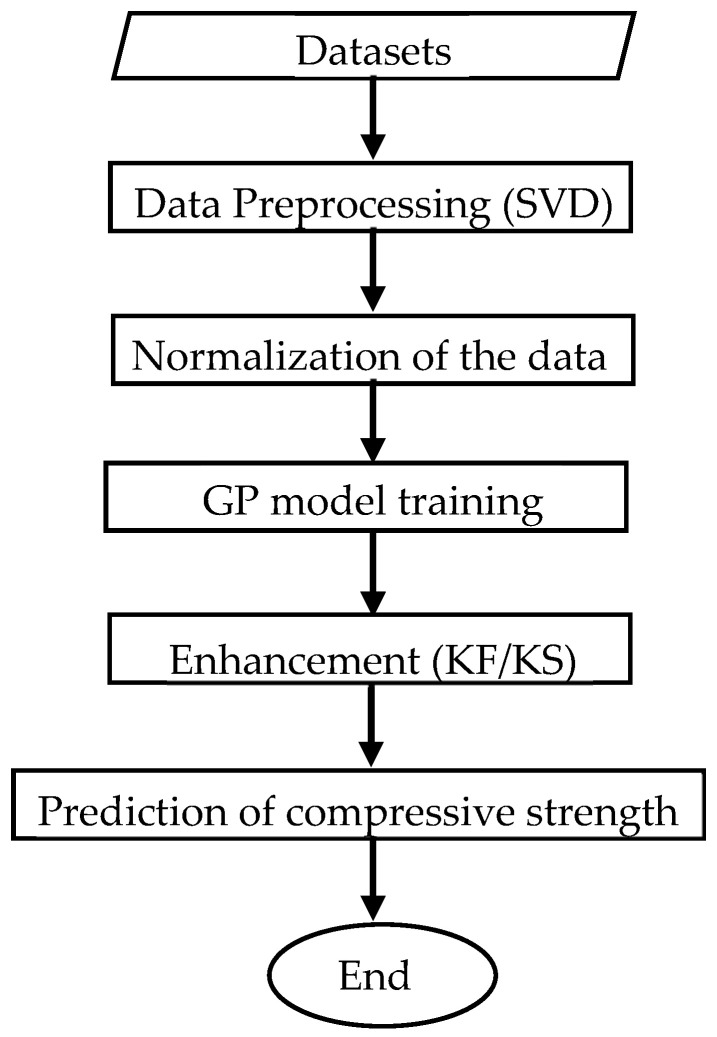
Flow chart of the framework.

**Figure 2 materials-17-06140-f002:**
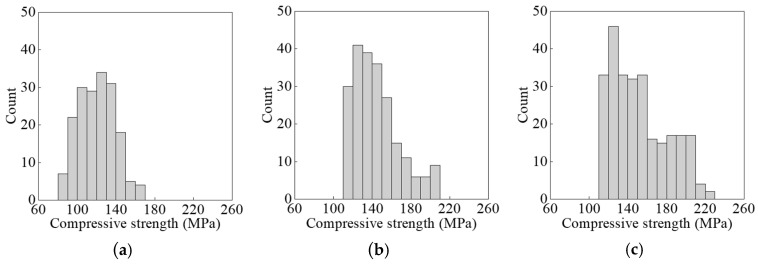
Distribution of the compressive strength: (**a**) Dataset 1; (**b**) Dataset 2; (**c**) Dataset 3.

**Figure 3 materials-17-06140-f003:**
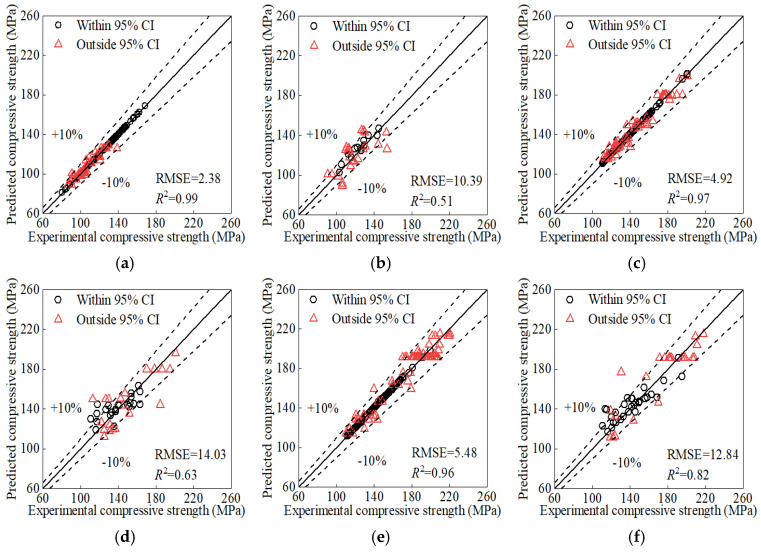
Prediction of three datasets (CI: Confidence Intervals): (**a**) training set of Dataset 1; (**b**) test set of Dataset 1; (**c**) training set of Dataset 2; (**d**) test set of Dataset 2; (**e**) training set of Dataset 3; (**f**) test set of Dataset 3.

**Figure 4 materials-17-06140-f004:**
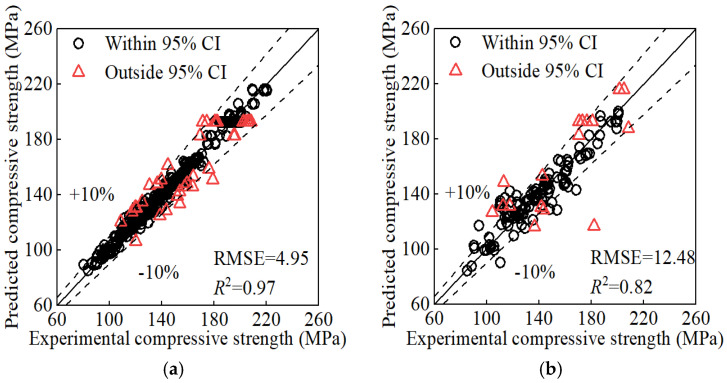
Prediction of the reconstructed input matrix: (**a**) training set; (**b**) test set.

**Figure 5 materials-17-06140-f005:**
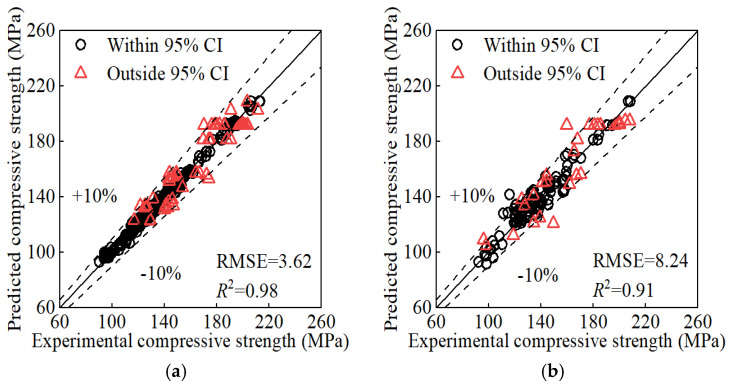
Prediction of KF/KS processed inputs: (**a**) training set; (**b**) test set.

**Figure 6 materials-17-06140-f006:**
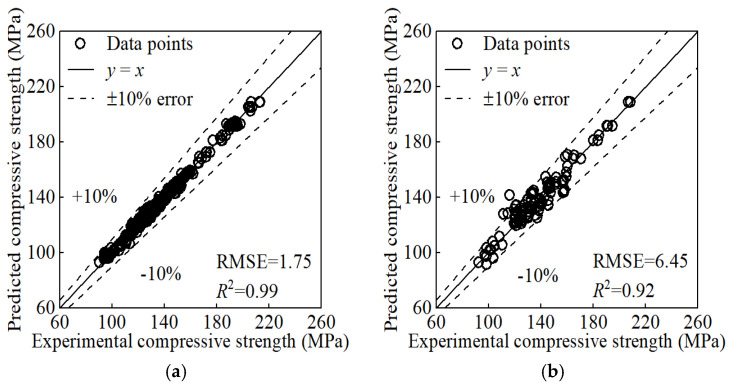
Prediction after high-uncertainty points had been removed: (**a**) training set; (**b**) test set.

**Figure 7 materials-17-06140-f007:**
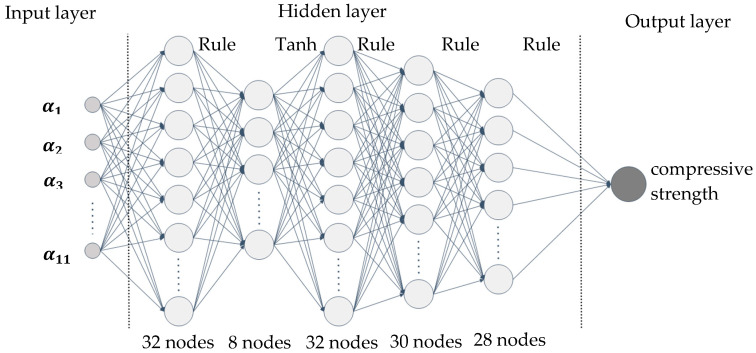
Structure of the ANN.

**Figure 8 materials-17-06140-f008:**
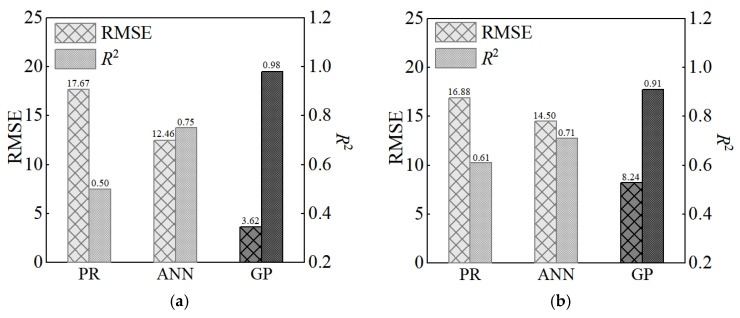
Comparison of prediction of the models: (**a**) training set; (**b**) test set.

**Figure 9 materials-17-06140-f009:**
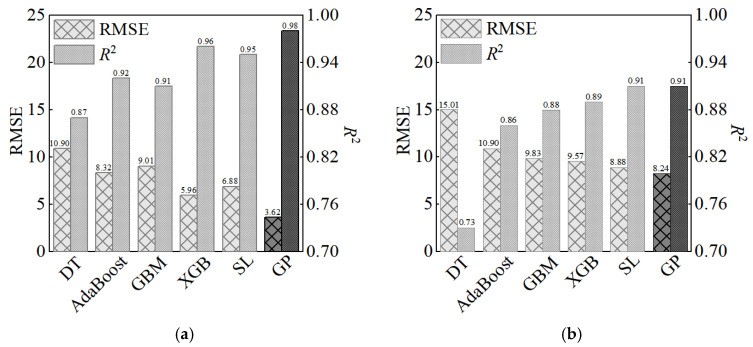
Prediction results in Reference [26] and those of the GP model: (**a**) training set; (**b**) test set.

**Figure 10 materials-17-06140-f010:**
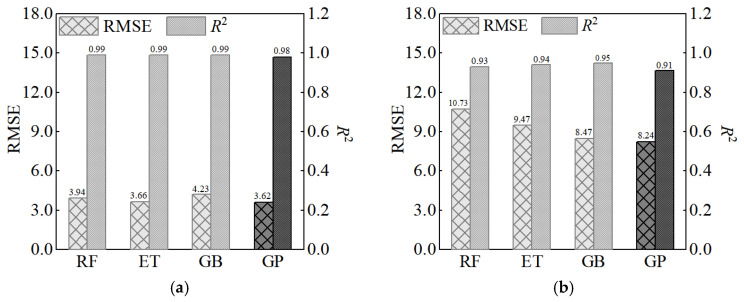
Prediction results in Reference [27] and those of the GP model: (**a**) training set; (**b**) test set.

**Table 1 materials-17-06140-t001:** Input variables.

	$\alpha_{1}$	$\alpha_{2}$	$\alpha_{3}$	$\alpha_{4}$	$\alpha_{5}$	$\alpha_{6}$	$\alpha_{7}$	$\alpha_{8}$	$\alpha_{9}$	$\alpha_{10}$	$\alpha_{11}$
Dataset 1											
Dataset 2											
Dataset 3											

Note: $\alpha_{1}$ denotes cement content; $\alpha_{2}$ denotes silica fume content; $\alpha_{3}$ denotes fly ash content; $\alpha_{4}$ denotes water content; $\alpha_{5}$ denotes blast furnace slag content; $\alpha_{6}$ denotes aggregate content; $\alpha_{7}$ denotes superplasticizer content; $\alpha_{8}$ denotes steel fiber content; $\alpha_{9}$ quartz powder content; $\alpha_{10}$ denote limestone powder content; $\alpha_{11}$ denotes nano-silica content.

**Table 2 materials-17-06140-t002:** Filling and merging of datasets.

	$\alpha_{1}$	$\alpha_{2}$	$\alpha_{3}$	$\alpha_{4}$	$\alpha_{5}$	$\alpha_{6}$	$\alpha_{7}$	$\alpha_{8}$	$\alpha_{9}$	$\alpha_{10}$	$\alpha_{11}$
Dataset 1									**[0]**	**[0]**	**[0]**
Dataset 2								**[0]**			**[0]**
Dataset 3											

## Data Availability

The original data presented in the study are available on request.

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
