# Peer review of "Enhanced Gaussian Process Model for Predicting Compressive Strength of Ultra-High-Performance Concrete (UHPC)"

_materials, 2024, doi:10.3390/ma17246140_

Round 1
Reviewer 1 Report
Comments and Suggestions for Authors
The manuscript is well written and properly organized and the results are clealry presented. Just a couple of minor suggestions:
1) the meaning of the acronym UHPC should also appear in the manuscript title.
2) the acronym PR should appear in the text accordingly to Figure 9.
3) Several typos and misspellings are present throughout the text.
Author Response
Thank you very much for taking the time to review this manuscript. We agree with your comments and have made the following changes:
1) the meaning of the acronym UHPC should also appear in the manuscript title.
Location: Revised manuscript(Page 1; Line 2)
Modification: Enhanced Gaussian Process Model for Predicting Compressive Strength of Ultra-High-Performance Concrete (UHPC)
2) the acronym PR should appear in the text accordingly to Figure 9.
Location: Revised manuscript(Page 12; Line 501)
Modification: Polynomial regression (PR) and artificial neural network (ANN) are……
3) Several typos and misspellings are present throughout the text.
Location: Revised manuscript(Page 3; Line 111)
Modification: Replace “enhance” with “enhances”
Revised text: it improves the prediction even with small datasets and enhances the interpretability
Location: Revised manuscript(Page 3; Line 112)
Modification: Replace “flexibly” with “flexibility”
Revised text: Third, the GP model demonstrates high flexibility by using different kernel functions
Location: Revised manuscript(Page 10; Line 416)
Modification: Add “to” before “reduce”
Revised text: Both KF and KS are based on Bayesian principles and fully utilize the available information to reduce the data dispersion
Location: Revised manuscript(Page 11; Line 461)
Modification: Replace “indicate” with “indicates”
Revised text: The resulting prediction Figure 5 indicates improvements compared to Figures 3 and 4
Location: Revised manuscript(Page 11; Line 464)
Modification: Remove “the” before “enhances”
Revised text: Additionally, it incorporates the physical law and enhances the reliability and interpretability of the prediction

Reviewer 2 Report
Comments and Suggestions for Authors
This paper applies a Gaussian process model combined with singular value decomposition and Kalman filtering and smoothing to predict the compressive strength of ultra-high-performance concrete based on the mix proportions.
The paper deals with an interesting topic, although some minor and major observations must be addressed before a final decision can be taken.
1.- The contributions and novelties of the paper are unclear. The final part of the Introduction section requires an extra effort to develop in detail this part.
2.- The Introduction sections must anticipate how the method proposed in this paper will be applied to real samples of concrete.
3.- Section 3.1 must be improved since it is not clear how the prediction is made.
4.- Figure 3. CI and RMSE have not been defined.
5.- Lines 288 to 293 are confused. Which is the relationship between f and F?
The Reviewer suggests revising the work based on the suggestions above in order to improve its readability, scientific interest and quality.
Author Response
Thank you very much for taking the time to review this manuscript. We agree with your comments and have made the following changes, more details in the word document.
1) The contributions and novelties of the paper are unclear. The final part of the Introduction section requires an extra effort to develop in detail this part.
Location: Revised manuscript(Page 3; Line 147)
Modification: Expand the final paragraph of the Introduction to clearly outline the contributions and novelties of the paper.
Revised text: Predicting the compressive strength of UHPC remains challenging due to the high variability of material properties and the limited availability of experimental data. Traditional methods often struggle to achieve a balance between accuracy, interpretability, and reliability. To address these limitations, this study presents three key contributions: (1) An enhanced Gaussian Process (GP) model is proposed, incorporating Singular Value Decomposition (SVD) and Kalman Filtering/Smoothing (KF/KS) to improve the prediction accuracy by increasing the volume of data and reducing data dispersion. (2) The GP model utilizes prior knowledge, including physical laws, to enhance prediction performance. (3) The model provides quantification of prediction uncertainty, which is advantageous for reliability assessment in engineering applications. These contributions aim to address the challenges in compressive strength prediction for UHPC, offering an efficient and reliable solution for optimizing UHPC mix designs.
2) The Introduction sections must anticipate how the method proposed in this paper will be applied to real samples of concrete.
Location: Revised manuscript(Page 3; Line 118)
Modification: Add a concise paragraph explaining the real-world application of the proposed model to UHPC, emphasizing its practical use in engineering.
Revised text: The GP model can be directly applied to the design of UHPC by predicting the compressive strength of new mix proportions without the need for physical testing. It can also provide predicted compressive strength values along with uncertainty estimates. This approach enables rapid evaluation and optimization of mix proportion designs, significantly reducing the time and cost of experiments while ensuring structural requirements are met in the early design stages
3) Section 3.1 must be improved since it is not clear how the prediction is made.
Location: Revised manuscript(Page 6; Line 220)
Modification: We have revised section 4.1 (In the original manuscript, in section 3.1) to ensure greater clarity
GP Model Introduction and Expression:
We began by introducing the GP model, defining its mathematical expression and explaining the key components—mean function and kernel function. This sets the stage for understanding the subsequent prediction steps.
Step-by-Step Prediction Process:
We have restructured Section 4.1 (In the original manuscript, in section 3.1) into a clear step-by-step process, which now includes:
Step 1: Definition of the model with a detailed explanation of the mean and kernel functions and their roles in incorporating prior knowledge.
Step 2: Training of the model by maximizing the log-likelihood function to optimize the hyperparameters.
Step 3: Setting up the joint distribution for the new data point and the existing training data.
Step 4: Calculating the posterior mean and variance for the new input, including the relevant equations.
Step 5: Generating the prediction, including the quantified uncertainty associated with the prediction.
Step 6: Updating the model as new data becomes available, emphasizing the adaptability of the GP model.
Enhanced Mathematical Explanation:
We included explicit references to the relevant equations in the model, ensuring that each step is mathematically grounded and that the reader can follow the prediction process clearly from training to inference.
Flow and Continuity:
The revised structure ensures a natural flow from model definition, through training, to making predictions, which makes the process easier to understand.
4) Figure 3. CI and RMSE have not been defined.
Location: Revised manuscript (Page 8; Line 345)
Modification: Add a brief explanation of CI in the figure caption to ensure immediate clarity for readers.
Revised text: Figure 3. Prediction of three datasets (CI: Confidence Intervals):
Location: Revised manuscript (Page 8; Line 329)
Modification: Add detailed definition and context for RMSE to clarify its role in evaluating the GP model's performance.
Revised text: During each cross-validation iteration, the performance metrics on the validation set were recorded, including the goodness-of-fit (denoted by R²) and root-mean-square error (RMSE). R² measures the consistency of the predicted values with the actual trend, and RMSE measures the overall deviation between predicted and actual values.
5) Lines 288 to 293 are confused. Which is the relationship between f and F?
Location: Revised manuscript (Page 10; Line 418)
Modification: Add clear explanations of f and F their respective definitions, and their roles in the prediction process. Simplify the presentation of their relationship in the context of the KF/KS while linking F to Abram’s formula for physical relevance.
Revised Text: 1) Calculate the prior prediction of the compressive strength for the current mix proportions: ft=Fft-1
where ft-1 is the posterior estimate of the compressive strength for the previous mix proportions; ft is the prior prediction of the compressive strength for the current mix proportions; F is the state transition matrix, which represents the ratio of the calculated results based on Abram's formula for the current and previous mix proportions: f=ax-b
Where f is the compressive strength of UHPC under specific mix proportions. x is the water-to-binder ratio; a, b are regression coefficients

Reviewer 3 Report
Comments and Suggestions for Authors
This paper proposes Prediction of the Compressive Strength for UHPC Based on Gaussian Process. The paper is well-written and provides valuable insights, but some concerns should be addressed:
• The authors should clarify the connections between the ultra-high-performance concrete (UHPC) based on its mix proportions and how each method contributes to the research. They should also provide a detailed explanation of the solution method used and its effectiveness.
- Please add a flowchart after the introduction.
-
Please add a literature review part after introduction.
-
Please mention the structure of your paper including sections and subsection in details.
- The title of the paper could be revised to better capture the essence and findings of the research, making it concise and engaging. A title that accurately reflects the key discoveries would be more suitable.
- The abstract should concisely outline the main findings in a well-structured, reader-friendly way, summarizing key insights to aid in understanding the research.
- The introduction should offer a well-organized framework for the research, ensuring smooth transitions between paragraphs. Improving coherence and linking paragraphs more effectively would strengthen the flow. Additionally, it should highlight the research's contributions, purpose, and significance by clearly stating the research question.
- To emphasize the study's relevance, the authors should reference recent related following papers published in MDPI journals such as ( Interpretable Machine Learning Models for Prediction of UHPC Creep Behavior. Buildings. 2024 Jul 7;14(7):2080), ( A novel machine learning approach combined with optimization models for eco-efficiency evaluation. Applied Sciences. 2020 Jul 28;10(15):5210.
- Figures should be presented with clarity and precision, with captions that provide sufficient context to aid reader comprehension.
- The research problem should be defined succinctly and clearly to enhance readability.
- Visual elements like figures and charts should be high quality, visually appealing, easy to interpret, and effectively convey the research findings. Figure 10, in particular, could benefit from additional data or more detailed information.
- The authors should offer more actionable insights for managers based on the findings, clarifying the practical applications for managerial and organizational use.
- Proofreading and careful editing are recommended to correct typographical and grammatical errors. An additional review by a fresh set of eyes may help catch any overlooked issues
The English could be improved to more clearly express the research.
Author Response
Thank you very much for taking the time to review this manuscript. We agree with your comments and have made the following changes, more details in the word document.
1) The authors should clarify the connections between the ultra-high-performance concrete (UHPC) based on its mix proportions and how each method contributes to the research. They should also provide a detailed explanation of the solution method used and its effectiveness.
Location:Revised manuscript (Page 1; Line 41)
Modification:Clarify the influence of UHPC mix proportions on compressive strength and emphasize the importance of predicting this relationship.
Revised Text: Ultra-high-performance concrete (UHPC) is characterized by its high strength, low porosity, and favorable ductility compared to normal concrete. These advantages have led to its increased application in engineering practice [1–3]. The compressive strength is one of the most fundamental mechanical properties of UHPC, which is critically influenced by the mix proportions, such as cement content, silica fume, fly ash, and the water-to-binder ratio. For instance, the water-to-binder ratio affects the hydration process, impacting strength development, while silica fume and fly ash refine the concrete microstructure, enhancing density and strength. Accurately predicting compressive strength based on these mix proportions is essential for optimizing UHPC performance. However, the prediction is challenging due to the complexity of the interactions among these components and their influence on strength.
Location:Revised manuscript (Page 9; Line 358)
Modification:Enhance the explanation of how SVD contributes to data preprocessing and its specific role in improving the GP model.
Revised Text: SVD extracts the principal components of the data, effectively filtering out less informative variations, while retaining the most critical features needed for accurate prediction. By simplifying the data while preserving its most essential features, SVD ensures that the GP model is trained on high-quality, well-structured data, ultimately enhancing the model's stability and reliability.
Location:Revised manuscript (Page 10; Line 407)
Modification:Improve the explanation of how KF/KS contribute to enhancing the GP model by incorporating physical constraints and improving prediction accuracy.
Revised Text: KF integrates prior physical knowledge directly into the predictive process to ensure the GP model’s prediction align with established physical principles. This step reduces data dispersion and refines the estimates by continuously updating predictions based on real-time observations. KS complements KF by applying a backward correction process to refine these predictions retrospectively, thus reducing residual errors and increasing overall consistency.
2) Please add a flowchart after the introduction.
Location: Revised manuscript (Page 4; Line 165)
Modification:Add a flowchart to visually summarize the research methodology after the introduction.
Revised Text: Please refer to the word file here
3) Please add a literature review part after introduction.
Location: Revised manuscript (Page 2; Line 51)
Modification: Thank you for the suggestion. The manuscript already includes an extensive discussion of relevant literature within the Introduction Section. This approach integrates the literature review into the problem context, ensuring a cohesive and logical presentation. Below, we highlight specific locations where key aspects of the literature are addressed:
- Experimental Approaches:
The second paragraph discusses experimental studies, such as Wang and Gao [4] on steel fiber content, Zhao et al. [5] on vibratory mixing, and Pourbaba et al. [6] on curing processes.
- Deterministic Models:
The third paragraph summarizes deterministic models, such as ensemble neural networks (Barkhordari et al. [9]), Random Forest combined with Firefly algorithms (Huang et al. [10]),
XGBoost and neural networks (Vargas et al. [13]).
- The fourth paragraph emphasizes the weaknesses of these approaches, particularly their lack of uncertainty quantification and integration with physical knowledge.
- Gaussian Process Model:
The fifth paragraph introduces the advantages of Gaussian Process (GP) models, including their ability to incorporate physical laws, provide probabilistic predictions, and handle small datasets.
These aspects set the foundation for the enhanced GP model proposed in this study.
4) Please mention the structure of your paper including sections and subsection in details.
Location: Revised manuscript (Page 3; Line 128)
Modification: The manuscript already includes a summary of the paper's structure in the Introduction Section. To provide a more detailed explanation, we have expanded the description to include subsections and the specific focus of each section.
Revised Text: The structure of the paper is as follows: Section 2 introduces the methodology. Section 3 introduces the datasets used to train and test the model. Section 4 describes the development and application of the GP model, covering its theoretical foundations, training process, hyperparameter optimization, and performance evaluation. Section 5 focuses on improving the GP model through data augmentation techniques, such as merging heterogeneous datasets using Singular Value Decomposition (SVD) and reducing data dispersion using Kalman Filtering and Smoothing (KF/KS). Section 6 provides a comprehensive comparison of the GP model’s predictions with alternative models, such as artificial neural networks (ANN) and regression-based methods, evaluating their relative accuracy and reliability. Finally, Section 7 concludes the study by summarizing the findings, highlighting the contributions, and discussing potential directions for future research.
5) The title of the paper could be revised to better capture the essence and findings of the research, making it concise and engaging. A title that accurately reflects the key discoveries would be more suitable.
Location: Revised manuscript (Page 1; Line 2)
Modification: Revise the title to make it concise, engaging, and representative of the research’s core contributions.
Revised Text: Enhanced Gaussian Process Model for Predicting Compressive Strength of Ultra-High-Performance Concrete (UHPC)
6) The abstract should concisely outline the main findings in a well-structured, reader-friendly way, summarizing key insights to aid in understanding the research.
Location: Revised manuscript (Page 1; Line 10)
Modification: Revise the abstract to provide a concise overview of the research problem, methodology, findings, and significance in a clear and structured format.
Revised Text: Ultra-high-performance concrete (UHPC) is widely used in engineering due to its exceptional mechanical properties, particularly compressive strength. Accurate prediction of the compressive strength is critical for optimizing mix proportions but remains challenging due to data dispersion, limited data availability, and complex material interactions. This study enhances the Gaussian Process (GP) model to address these challenges by incorporating Singular Value Decomposition (SVD) and Kalman Filtering/Smoothing (KF/KS). SVD improves data quality by extracting critical features, while KF/KS reduces data dispersion and align prediction with physical laws. The enhanced GP model predicts compressive strength with improved accuracy and quantifies uncertainty, offering significant advantages over traditional methods. Results demonstrate that the enhanced GP model outperforms other models, including artificial neural networks (ANN) and regression models, in terms of reliability and interpretability. This approach provides a robust tool for optimizing UHPC mix designs, reducing experimental costs, and ensuring structural performance.
7) The introduction should offer a well-organized framework for the research, ensuring smooth transitions between paragraphs. Improving coherence and linking paragraphs more effectively would strengthen the flow. Additionally, it should highlight the research's contributions, purpose, and significance by clearly stating the research question.
Location: Revised manuscript (Page 1; Line 35)
Modification:
The introduction has been revised with the following improvements:
- Coherence:
Smooth transitions have been added between paragraphs to link discussions about experimental methods, deterministic models, and machine learning techniques with the research's focus on Gaussian Process (GP) models.
- Clarity of Research Purpose and Contributions:
The research question is explicitly stated, and the key contributions are summarized clearly.
- Framework for the Paper:
A detailed description of the paper's structure has been provided to guide readers through the study.
Key Improvements in the Revised Text:
- Linking Experimental and Predictive Approaches:
Experimental methods provide valuable insights into UHPC compressive strength but are time-intensive and resource-heavy, highlighting the need for efficient predictive models. Deterministic and machine learning models offer solutions but lack the ability to quantify uncertainty or incorporate prior knowledge, making them less reliable for real-world applications.
- Establishing the Need for GP Models:
The Gaussian Process (GP) model addresses these limitations by offering probabilistic predictions, incorporating physical laws, and handling small datasets effectively. This makes it an ideal candidate for predicting UHPC compressive strength.
- Smooth Transitions:
Each paragraph builds logically on the previous one, e.g., transitioning from limitations of experimental methods to the need for advanced predictive models, then to the advantages of GP models.
- Clear Research Contributions:
The key contributions of this study are threefold: (1) We propose an enhanced GP model that incorporates SVD and KF/KS to improve prediction accuracy of discrete UHPC data. (2) We utilize prior knowledge, including physical laws, to improve prediction performance. (3) The model provides quantification of prediction uncertainty, advantageous for reliability assessment in engineering applications.
- Paper Framework:
The structure of the paper is as follows: In Section 2, we introduce the methodology...Finally, Section 7 concludes the study by summarizing the findings, highlighting the contributions, and discussing potential directions for future research.
8) To emphasize the study's relevance, the authors should reference recent related following papers published in MDPI journals such as ( Interpretable Machine Learning Models for Prediction of UHPC Creep Behavior. Buildings. 2024 Jul 7;14(7):2080), ( A novel machine learning approach combined with optimization models for eco-efficiency evaluation. Applied Sciences. 2020 Jul 28;10(15):5210.
Location: Revised manuscript (Page 2; Line 89)
Modification: We have referenced the two papers provided to emphasize the relevance of the study
Revised Text: Recent studies have demonstrated the potential of machine learning approaches in advancing UHPC-related research and concrete optimization. For example, interpretable machine learning models have been used to predict UHPC creep behavior, showcasing their ability to balance accuracy and transparency while addressing complex mechanical behaviors [14]. Similarly, machine learning approaches combined with optimization models have been applied to evaluate eco-efficiency, enabling sustainable design in concrete applications [15].
9) Figures should be presented with clarity and precision, with captions that provide sufficient context to aid reader comprehension.
Location: Revised manuscript (All the Figures)
Modification: Thank you for the valuable suggestion. We have carefully reviewed all figures in the manuscript to ensure their clarity and precision. Additionally, figure captions have been revised to provide sufficient context, explaining the content and purpose of each figure in a concise yet detailed manner.
Revised text:
- Figure Quality:
All figures have been checked to ensure they are of high resolution and appropriately labeled for clarity. Graphical elements such as legends, axes, and text have been adjusted for better visibility and readability. This ensures that figures are visually accessible and effectively convey their intended information.
- Figure Captions:
Captions have been rewritten to provide detailed context, explaining what each figure represents, the methodology used (if applicable), and how it contributes to the study.
10) The research problem should be defined succinctly and clearly to enhance readability.
Location: Revised manuscript (Page 2; Line 51)
Modification: The second paragraph: Discusses the limitations of traditional experimental methods.
Revised text: However, curing and testing consume too much time and yield limited results.
Location: Revised manuscript (Page 3; Line 99)
Modification: The fourth paragraph: Highlights the challenges with existing predictive models.
Revised text: However, these models cannot represent the uncertainty of the compressive strength of the UHPC [16–18]. The inherent uncertainty results from the variability of ingredient in the mix, diversity of curing environment, difference in hydration process, and so on.
11) Visual elements like figures and charts should be high quality, visually appealing, easy to interpret, and effectively convey the research findings. Figure 10, in particular, could benefit from additional data or more detailed information.
Location: Revised manuscript (Page 14; Line 546)
Modification:
- Improved Visual Quality:
Increased the resolution of Figure 9 (In the original manuscript, Figure 10) to ensure high-quality graphics. The elements such as axes labels, bar textures, and numbers have been adjusted to ensure all elements are visible and easily distinguishable. Enhancing the figure’s quality improves readability, particularly for viewers who may need to examine the chart in detail.
- Additional Data and Clarity in Visualization:
Changes to Figure Elements:
Added Numerical Values Above Bars: The RMSE and R2 values have been placed directly above each corresponding bar to make it easier for readers to interpret the figure without having to match the legend.
Distinct Color Scheme and Labels: Updated the visual style by using different, more distinguishable patterns or colors for the bars representing RMSE and R2. The updated colors ensure that readers can easily differentiate between RMSE and R2 metrics, even when printed in grayscale.
Labeled Axes More Clearly: Adjusted the left y-axis to explicitly indicate "RMSE" and the right y-axis to show "R2" for easier interpretation. This dual-axis setup provides clear information about what each bar represents without ambiguity.
12) The authors should offer more actionable insights for managers based on the findings, clarifying the practical applications for managerial and organizational use.
Location: Revised manuscript (Page 15; Line 570)
Modification: Added a paragraph to provide actionable insights for managers.
Revised Text: The enhanced GP model enables managers to efficiently evaluate mix proportions, significantly reducing costs and time compared to traditional physical testing. Additionally, the model's ability to quantify prediction uncertainty supports more informed decisions in quality control and risk management for UHPC production.
13) Proofreading and careful editing are recommended to correct typographical and grammatical errors. An additional review by a fresh set of eyes may help catch any overlooked issues
Comments on the Quality of English Language
The English could be improved to more clearly express the research.
Proofreading: A detailed review was conducted to identify and correct typographical errors, grammatical issues, and awkward phrasing.
Location: Revised manuscript(Page 3; Line 111)
Original: “it improves the prediction even with small datasets and enhance the interpretability”
Modification: Replace “enhance” with “enhances”
Revised text: “it improves the prediction even with small datasets and enhances the interpretability”
Location: Revised manuscript(Page 3; Line 112)
Original: “Third, the GP model demonstrates high flexibly by using different kernel functions”
Modification: Replace “flexibly” with “flexibility”
Revised text: “Third, the GP model demonstrates high flexibility by using different kernel functions”
Location: Revised manuscript(Page 10; Line 416)
Original: “Both KF and KS are based on Bayesian principles and fully utilize the available information reduce the data dispersion”
Modification: Add “to” before “reduce”
Revised text: “Both KF and KS are based on Bayesian principles and fully utilize the available information to reduce the data dispersion”
Location: Revised manuscript(Page 11; Line 461)
Original: “The resulting prediction Figure 6 indicate improvements compared to Figures 3 and 5”
Modification: Replace “indicate” with “indicates”
Revised text: “The resulting prediction Figure 5 indicates improvements compared to Figures 3 and 4”
Location: Revised manuscript(Page 11; Line 464)
Original: “Additionally, it incorporates the physical law and the enhances the reliability and interpretability of the prediction”
Modification: Remove “the” before “enhances”
Revised text: “Additionally, it incorporates the physical law and enhances the reliability and interpretability of the prediction”

Reviewer 4 Report
Comments and Suggestions for Authors
Review
Here are my conclusions and observations after reviewing the article entitled “Prediction of the Compressive Strength for UHPC Based on Gaussian Process”.
The main contribution of the article is the application of an improved Gaussian process (GP) model to predict the compressive strength of ultra-high performance concrete (UHPC), addressing the inherent uncertainty in the experimental data. By integrating singular value decomposition (SVD) techniques to increase the data volume and Kalman filtering (KF/KS) to reduce dispersion, the GP model offers an accurate and reliable prediction, outperforming previous deterministic methods and machine learning models in terms of generalization and uncertainty quantification. The article is well written and organized in a logical and sequential manner, lending itself to a fluid and well-supported reading with references to figures and equations.
Abstract
The abstract is well written and summarizes the main ideas of the research, as well as the main findings of the authors. Introduction
The introduction has been well done, presenting references to relevant works in the area, all of them contributing value to the subject developed by the authors, as well as having updated and recognized references. Please unify references 17, 18 and 19.
Methodology
The methodology is briefly introduced at the end of the introduction. This section directly indicates the parts of the article, however, the methodology to be applied, the objectives and the scope of the research are not explicitly stated. Then, in the section, the experimental components are described directly. I consider it convenient that the authors dedicate the beginning of section two to describing the methodology in a precise manner.
Regarding the study and its parts, I have some observations, which I consider important that the authors can justify or include in the discussion of their results, since these aspects can influence the results obtained.
The article uses a linear normalization of the input variables before training the model, which can be limited given that the distribution of the data is not always linear. This can affect the predictive accuracy and generalization of the model in data sets with non-linear distributions.
The process of combining different data sets using singular value decomposition (SVD) simplifies the data, but by introducing zero values into missing columns of certain sets, it is assumed that these variables do not influence, which can be incorrect and strongly affect the integrity of the data and the robustness of the model.
The use of Kalman filters (KF) and Kalman smoothing (KS) to reduce the dispersion in the data could limit the interpretation of the natural variability in the experimental results. This approach can reduce the ability of the model to capture real fluctuations, especially if the data have high heterogeneity.Se selecciona una función kernel Matern 5/2 sin una justificación sólida sobre por qué esta opción es la óptima frente a otras funciones kernel.
The strategy of removing data points with high uncertainty, based on model confidence intervals, may bias the model, excluding valid cases and reducing generalizability in future predictions.
About the results and conclusions
Overall, the conclusions are well-founded in the results obtained. However, some aspects, such as the exclusion of high uncertainty data and the selection of the kernel function, could be discussed in more depth to give a complete overview of the model's limitations in practical scenarios. This would help improve the robustness of the conclusions by addressing potential biases and the specific methodological choice.
References
As I indicated above, the references are appropriate for the topic of the article. I have also reviewed the format of the references and it is in line with that required by the journal.
Typing and grammatical errors
In Equation 1, avoid using bold in the definition of variables. The same applies to Equation 2 for terms that are not vectors or matrices. Observe the same notation from now on with the rest of the equations.
Some grammatical errors were detected during the reading of the manuscript. In any case, the authors are suggested to carry out a thorough review of the entire manuscript.
Page 2, Line 89: "including plausible physical low" Correction: "low" should be "law" — "including plausible physical law."
Page 3, Line 111: "These data cover a broad range of UHPC mixtures and reflect a proper range of compressive strength." Suggested correction: Please, consider rephrasing for clarity: "These data cover a broad range of UHPC mixtures and compressive strength values."
Page 2, Line 114: "Dataset 1 were our experimental results" Here "were" should be "was" — "Dataset 1 was our experimental results." Dataset is singular.
Page 13, Line 270: "it shows obvious improvement in predictive performance." I suggest that "obvious" should be "an obvious" — "it shows an obvious improvement in predictive performance."
Page 8, Line 327: "and the enhances the reliability and interpretability of the prediction." Consider to remove "the" — "and enhances the reliability and interpretability of the prediction."
Page 9, Line 384: "whereas the proposed method required only 459.30 seconds." I suggest the following improvement: Change "required" to "requires" for consistency — "whereas the proposed method requires only 459.30 seconds."
Author Response
Thank you very much for taking the time to review this manuscript. We agree with your comments and have made the following changes, more details in the word document.
Abstract
The abstract is well written and summarizes the main ideas of the research, as well as the main findings of the authors. Introduction
The introduction has been well done, presenting references to relevant works in the area, all of them contributing value to the subject developed by the authors, as well as having updated and recognized references. Please unify references 17, 18 and 19.
Location: Revised manuscript(Page 3; Line 111)
Modification: Combined references 17, 18, and 19 into a single reference, as they all support the same point regarding Bayesian principles and the benefits of the Gaussian Process model.
Revised Text: “According to the Bayesian principles, it improves the prediction even with small datasets and enhances the interpretability[19-21].”
Methodology
The methodology is briefly introduced at the end of the introduction. This section directly indicates the parts of the article, however, the methodology to be applied, the objectives and the scope of the research are not explicitly stated. Then, in the section, the experimental components are described directly. I consider it convenient that the authors dedicate the beginning of section two to describing the methodology in a precise manner.
Location: Revised manuscript(Page 4; Line 166)
Modification: Added methodology as Section 2:
Revised Text: “In this section, we present the methodology adopted in this research to predict the compressive strength of Ultra-High-Performance Concrete (UHPC). The research objectives are to (1) develop an enhanced Gaussian Process (GP) model that accurately predicts compressive strength based on mix proportions, (2) incorporate data augmentation techniques such as Singular Value Decomposition (SVD) and Kalman Filtering/Smoothing (KF/KS) to improve model reliability, and (3) provide uncertainty quantification to support practical decision-making. The scope of this research encompasses data preprocessing, model training, and evaluation using experimental datasets of UHPC with various mix compositions. The methodology involves collecting datasets, preprocessing using SVD to manage data heterogeneity, and using KF/KS to improve training data quality. The enhanced GP model is trained and validated against other predictive models, such as ANN and Polynomial Regression (PR), to demonstrate its effectiveness in compressive strength prediction.”
Regarding the study and its parts, I have some observations, which I consider important that the authors can justify or include in the discussion of their results, since these aspects can influence the results obtained.
The article uses a linear normalization of the input variables before training the model, which can be limited given that the distribution of the data is not always linear. This can affect the predictive accuracy and generalization of the model in data sets with non-linear distributions.
Location: Revised manuscript(Page 5; Line 207)
Modification: Clarified the normalization method used in the study and acknowledged potential limitations in dealing with non-linear datasets.
Revised Text: In this study, Min-Max normalization was used to preprocess the input variables before training the model. This approach scales each feature to a range of [0, 1], preserving the relative relationships within the data, which helps manage different variable scales effectively.
The process of combining different data sets using singular value decomposition (SVD) simplifies the data, but by introducing zero values into missing columns of certain sets, it is assumed that these variables do not influence, which can be incorrect and strongly affect the integrity of the data and the robustness of the model.
Location: Revised manuscript(Page 9; Line 371)
Modification: Thank you for pointing out this important aspect regarding the handling of missing values during the SVD process. We acknowledge that introducing zeros into missing columns may imply that these variables have no influence, which can potentially affect data integrity and model robustness, especially when the original data contains meaningful missing patterns.
Revised Text: Currently, we opted for zero-value imputation as an initial approach to ensure consistency across the combined datasets and facilitate the Singular Value Decomposition process. We recognize that this may not be the most optimal approach in all cases. Moving forward, we plan to improve our handling of missing data by considering alternative imputation methods, such as using mean or median values, as well as exploring advanced techniques like matrix factorization and machine learning-based imputation. These improvements will aim to enhance the reliability of the model and ensure that the impact of missing data is adequately addressed.
The use of Kalman filters (KF) and Kalman smoothing (KS) to reduce the dispersion in the data could limit the interpretation of the natural variability in the experimental results. This approach can reduce the ability of the model to capture real fluctuations, especially if the data have high heterogeneity. The selection of the Matern 5/2 kernel needs a solid justification on why this option is optimal compared to other kernel functions.
Location: Revised manuscript(Page 11; Line 465)
Modification: Adding the discussion about potential limitations and demonstrates that the potential negative impact of KF/KS was considered and mitigated.
Revised Text: Using KF/KS to reduce data dispersion might limit the interpretation of natural variability, particularly for datasets characterized by high heterogeneity. We carefully chose KF/KS as these techniques effectively reduce measurement noise. This step aimed to provide more stable and reliable predictions. We understand the importance of retaining the natural variability in data, and in future studies, we plan to conduct detailed sensitivity analyses to evaluate the impact of KF/KS more thoroughly.
Location: Revised manuscript(Page 7; Line 309)
Modification: Providing a direct comparison with other kernel functions makes the rationale for selecting Matern 5/2 much clearer.
Revised Text: We selected the Matern 5/2 kernel function due to its flexibility in handling variations across different scales, which is crucial for UHPC data characterized by heterogeneous and non-linear relationships. Compared to other potential kernels, such as the Radial Basis Function (RBF) kernel or the Matern 3/2 kernel, the Matern 5/2 kernel strikes an ideal balance between smoothness and adaptability. The RBF kernel, while providing very smooth predictions, can be too restrictive and over-smooth complex real-world data, potentially missing important local variations. On the other hand, the Matern 3/2 kernel offers less smoothness but does not capture the moderate variability as effectively as the Matern 5/2 kernel. Empirical testing demonstrated that the Matern 5/2 kernel consistently provided better predictive accuracy and generalization capability, making it the optimal choice for modeling the compressive strength of UHPC.
The strategy of removing data points with high uncertainty, based on model confidence intervals, may bias the model, excluding valid cases and reducing generalizability in future predictions.
Location: Revised manuscript(Page 11; Line 482)
Modification: Thank you for pointing out this important concern. We understand that removing data points with high uncertainty, as determined by model confidence intervals, could potentially introduce bias, leading to a reduction in the generalizability of the model. Currently, the decision to remove high-uncertainty data points was made to enhance model stability and reduce the influence of potential outliers or highly noisy data that could distort the predictions. We recognize that this approach may result in excluding data points that represent the inherent variability of UHPC compressive strength, which is important for a model that aims to generalize well to diverse datasets. Moving forward, if an opportunity arises, we plan to explore alternative strategies, such as applying weighted training methods where high-uncertainty data points are retained but given lower weights during training. This could help preserve the diversity of the dataset while minimizing the influence of noisy data. However, given the scope of the current study, we opted for the more straightforward approach of removing high-uncertainty points to ensure model reliability and reduce the effects of significant noise.
Revised Text: This approach may introduce bias by excluding potentially valid data points, which could limit the model's ability to generalize well to new datasets. If feasible in future work, we aim to explore alternative approaches, such as using a weighted training scheme where data points with higher uncertainty are retained but assigned lower weights. This could allow us to better preserve data variability while minimizing the negative impact of noisy data on the model’s performance. However, given the current scope of this research, the focus was on ensuring model reliability and mitigating noise effects.
About the results and conclusions
Overall, the conclusions are well-founded in the results obtained. However, some aspects, such as the exclusion of high uncertainty data and the selection of the kernel function, could be discussed in more depth to give a complete overview of the model's limitations in practical scenarios. This would help improve the robustness of the conclusions by addressing potential biases and the specific methodological choice.
Location: Revised manuscript(Page 15; Line 576)
Modification: To make the conclusion section more precise, coherent, and reflective of the key contributions and limitations of the study, we emphasize the strengths while also acknowledge the limitations brought up by the reviewers.
Revised Text: It is important to recognize certain limitations. The exclusion of data points with high uncertainty may introduce bias, potentially affecting generalizability. Future work should explore weighting schemes instead of removing uncertain data entirely, to better preserve data diversity. Furthermore, the choice of the Matern 5/2 kernel for the GP model was effective for the given dataset, but alternative kernels should be explored in future research to confirm the robustness of this choice across various data scenarios.
References
As I indicated above, the references are appropriate for the topic of the article. I have also reviewed the format of the references and it is in line with that required by the journal.
Typing and grammatical errors
In Equation 1, avoid using bold in the definition of variables. The same applies to Equation 2 for terms that are not vectors or matrices. Observe the same notation from now on with the rest of the equations.
Thank you for pointing out the inconsistencies in the formatting of the variables in our equations. We have reviewed the notation used in all equations and made appropriate changes to ensure consistency throughout the manuscript. Below, we provide a detailed summary of our modifications:
- Equation 1 (Normalization Formula):
Location: Revised manuscript(Page 5; Line 212)
Modification: After careful review, we determined that in Equation 1, the variables representing input values for Min-Max normalization are indeed vectors, and therefore the bold formatting is appropriate.
Revised Text: Please refer to the word file here
- Equation 2 (Gaussian Process Mean Function):
Location: Revised manuscript(Page 6; Line 223)
Modification: We revised the notation to consistently use bold only for vectors and matrices.
Revised Text: Please refer to the word file here
- General Consistency for All Equations:
We reviewed the entire manuscript to ensure consistent formatting for vectors and scalars across all equations.
Some grammatical errors were detected during the reading of the manuscript. In any case, the authors are suggested to carry out a thorough review of the entire manuscript.
Page 2, Line 89: "including plausible physical low" Correction: "low" should be "law" — "including plausible physical law."
Location: Revised manuscript(Page 3; Line 109)
Original Text: including plausible physical low
Revised Text: including plausible physical law
Page 3, Line 111: "These data cover a broad range of UHPC mixtures and reflect a proper range of compressive strength." Suggested correction: Please, consider rephrasing for clarity: "These data cover a broad range of UHPC mixtures and compressive strength values."
Location: Revised manuscript(Page 5; Line 187)
Original Text: These data cover a broad range of UHPC mixtures and reflect a proper range of compressive strength.
Revised Text: These data cover a broad range of UHPC mixtures and compressive strength values.
Page 2, Line 114: "Dataset 1 were our experimental results" Here "were" should be "was" — "Dataset 1 was our experimental results." Dataset is singular.
Location: Revised manuscript(Page 5; Line 188)
Original Text: Dataset 1 were our experimental results
Revised Text: Dataset 1 consists of experimental results obtained in this study
Page 13, Line 270: "it shows obvious improvement in predictive performance." I suggest that "obvious" should be "an obvious" — "it shows an obvious improvement in predictive performance."
Location: Revised manuscript(Page 10; Line 399)
Original Text: it shows obvious improvement in predictive performance
Revised Text: it shows an obvious improvement in predictive performance
Page 8, Line 327: "and the enhances the reliability and interpretability of the prediction." Consider to remove "the" — "and enhances the reliability and interpretability of the prediction."
Location: Revised manuscript(Page 11; Line 464)
Original Text: and the enhances the reliability and interpretability of the prediction
Revised Text: and enhances the reliability and interpretability of the prediction
Page 9, Line 384: "whereas the proposed method required only 459.30 seconds." I suggest the following improvement: Change "required" to "requires" for consistency — "whereas the proposed method requires only 459.30 seconds."
Location: Revised manuscript(Page 13; Line 512)
Thank you for pointing out the need for consistency in tense usage. After reviewing the context of the surrounding text, we believe that maintaining the past tense ("required") is appropriate in this instance. The comparison in this section is made in the context of an experiment that has already been conducted, and therefore the past tense better reflects that these results were obtained during a specific instance of model evaluation.

Reviewer 5 Report
Comments and Suggestions for Authors
Dear Authors
The work needs to be improved before acceptance, and I suggest major revision. I hope that the given comments will be useful for improving the quality of the work.
Review:
Title: Prediction of the Compressive Strength for UHPC Based on Gaussian Process
1) I don't know if the authors used a template for writing the paper because it is not visible when the paper is downloaded;
2) In the last paragraph of the introductory part of the work, it is necessary to avoid writing in the first person and clearly write what the goal of the work is and not to look like a report;
3) It is necessary to give at least some basic characteristics of the UHPC used in the work;
4) Lines 114, 196 etc. - avoid writing in the first person;
5) Figure 1 is not the clearest; Show it more clearly or in a table;
Figures 3, 5, and 7 nothing is clearly visible. Make different symbols at least in some colors;
Author Response
Thank you very much for taking the time to review this manuscript. We agree with your comments and have made the following changes, more details in the word document.
1) I don't know if the authors used a template for writing the paper because it is not visible when the paper is downloaded;
Template Application: We have re-applied the official template to the manuscript to guarantee that all formatting, including fonts, headings, and layout, aligns with the journal's standards. We have also verified that the newly updated manuscript now maintains the intended formatting throughout the submission process, ensuring visibility when downloaded.
2) In the last paragraph of the introductory part of the work, it is necessary to avoid writing in the first person and clearly write what the goal of the work is and not to look like a report;
Location: Last paragraph of the introductory part of the work (Page 3; Line 126)
Modification: Avoid the use of the first person and clearly state the goal of the work. Reframe the paragraph to focus on the objectives and structure of the paper, rather than sounding like a report.
Revised Text: Considering the advantages of the GP model, this study seeks to apply it to predict the compressive strength of UHPC. The structure of the paper is as follows: Section 2 introduces the methodology. Section 3 introduces the datasets used to train and test the model. Section 4 describes the development and application of the GP model, covering its theoretical foundations, training process, hyperparameter optimization, and performance evaluation. Section 5 focuses on improving the GP model through data augmentation techniques, such as merging heterogeneous datasets using Singular Value Decomposition (SVD) and reducing data dispersion using Kalman Filtering and Smoothing (KF/KS). Section 6 provides a comprehensive comparison of the GP model’s predictions with alternative models, such as artificial neural networks (ANN) and regression-based methods, evaluating their relative accuracy and reliability. Finally, Section 7 concludes the study by summarizing the findings, highlighting the contributions, and discussing potential directions for future research. Figure 1 provides an overview of the research methodology, outlining the key steps from data collection and preprocessing to model training, enhancement, and final prediction.
3) It is necessary to give at least some basic characteristics of the UHPC used in the work;
Location: Revised manuscript(Page 1; Line 41)
Modification: We have included information on the composition of UHPC (e.g., high cement content, silica fume, fly ash, and superplasticizers) and its key properties (e.g., compressive strength range and low porosity). This addition ensures that readers understand the nature of the UHPC used in the study, which provides valuable context for interpreting the experimental data and results.
Revised Text: which is critically influenced by the mix proportions, such as cement content, silica fume, fly ash, and the water-to-binder ratio. For instance, the water-to-binder ratio affects the hydration process, impacting strength development, while silica fume and fly ash refine the concrete microstructure, enhancing density and strength. Accurately predicting compressive strength based on these mix proportions is essential for optimizing UHPC performance. However, the prediction is challenging due to the complexity of the interactions among these components and their influence on strength.
4) Lines 114, 196 etc. - avoid writing in the first person;
Location: Revised manuscript(Page 5; Line 188)
Original Text: Dataset 1 was our experimental results
Revised Text: Dataset 1 consists of experimental results obtained in this study.
Location: Revised manuscript(Page 7; Line 306)
Original Text: We selected the mean of the training set as the prior mean function for the GP model
Revised Text: The mean of the training set was selected as the prior mean function for the GP model.
5) Figure 1 is not the clearest; Show it more clearly or in a table;
Location: Revised manuscript(Page 5; Line 196)
Modification: We have replaced Figure 1 with a new Table 1 that presents the input variables for each dataset in a clear and structured tabular format. This modification enhances readability and makes it easier for readers to interpret the input variables used in each dataset.
Revised Text:
|
|||||||||||
Dataset 1 |
|
|
|
||||||||
Dataset 2 |
|
|
|||||||||
Dataset 3 |
6) Figures 3, 5, and 7 nothing is clearly visible. Make different symbols at least in some colors;
Location: Revised manuscript(Page 8; Line 345、Page 10; Line 401、Page 12; Line 495)
Modification: The datasets are now differentiated using black circles and red triangles. The symbols are distinctly colored to help readers easily distinguish between different datasets.

Round 2
Reviewer 2 Report
Comments and Suggestions for Authors
The authors have replied all my questions
Reviewer 3 Report
Comments and Suggestions for Authors
Authors answered all my comments and this version is acceptable.
Comments on the Quality of English LanguageThe English could be improved to more clearly express the research.
Reviewer 4 Report
Comments and Suggestions for Authors
I have no additional comments.
Reviewer 5 Report
Comments and Suggestions for Authors
Daer Authors,
The authors have now corrected the paper, and it can be accepted. Only let them check once more whether it was done in Template, because it doesn't look like that to me.